# Wireless Power Transfer Efficiency Optimization Tracking Method Based on Full Current Mode Impedance Matching

**DOI:** 10.3390/s24092917

**Published:** 2024-05-02

**Authors:** Yuanzhong Xu, Yuxuan Zhang, Tiezhou Wu

**Affiliations:** Hubei Key Laboratory for High-Efficiency Utilization of Solar Energy and Operation Control of Energy Storage System, Hubei University of Technology, Wuhan 430063, China; xuyuanzhong@hbut.edu.cn (Y.X.); 19881031@hbut.edu.cn (T.W.)

**Keywords:** wireless power transfer, full current mode, efficiency tracking, efficiency detection, MATLAB simulation

## Abstract

Wireless power transfer (WPT) technology is a contactless wireless energy transfer method with wide-ranging applications in fields such as smart homes, the Internet of Things (IoT), and electric vehicles. Achieving optimal efficiency in wireless power transfer systems has been a key research focus. In this paper, we propose a tracking method based on full current mode impedance matching for optimizing wireless power transfer efficiency. This method enables efficiency tracking in WPT systems and seamless switching between continuous conduction mode and discontinuous mode, expanding the detection capabilities of the wireless power transfer system. MATLAB was used to simulate the proposed method and validate its feasibility and effectiveness. Based on the simulation results, the proposed method ensures optimal efficiency tracking in wireless power transfer systems while extending detection capabilities, offering practical value and potential for widespread applications.

## 1. Introduction

Wireless power transfer (WPT) technology is an emerging wireless energy transfer method that offers high efficiency and reliability, with wide-ranging applications in energy, electric vehicle charging, biomedical implants, etc. [1,2,3,4,5]. However, achieving optimal efficiency in wireless power transfer systems remains a significant challenge [6,7,8]. Existing control methods for WPT systems have certain limitations and issues, such as low transmission efficiency, short transmission distances, and high system costs [9,10]. Therefore, developing new control methods to address these challenges is a current research focus in wireless power transfer technology [11,12].

In the field of tracking and detection for wireless power transfer systems, active impedance matching and passive impedance matching are two key techniques for achieving optimal efficiency tracking in wireless power transfer systems. These techniques adjust circuit parameters between the transmitter and receiver to achieve impedance matching and maximize power transmission.

Active impedance matching adds active components to adjust circuit parameters and achieves impedance matching. An efficiency optimization method based on active impedance matching was proposed in [13] to enhance power transmission efficiency by adjusting the impedance matching circuit between the transmitter and receiver. Similarly, ref. [14] presents an adaptive power transmission method based on active impedance matching for different load conditions. These methods typically utilize components such as transformers, switch circuits, and amplifiers to adjust impedance matching. However, active impedance matching techniques have limitations and challenges. Firstly, they often require additional active components, increasing system costs and complexity. Moreover, factors such as temperature and noise may affect these active components, leading to system instability [15]. Additionally, active impedance matching methods require a trade-off between power loss and system stability while pursuing power transmission efficiency [16].

By contrast, passive impedance matching achieves impedance matching by adjusting circuit component parameters, such as inductors, capacitors, and resistors, between the transmitter and receiver. Resonant circuits are commonly used as passive impedance matching elements, which adjust parameters such as inductance and capacitance to achieve impedance matching between the transmitter and receiver. Compensation circuits, another passive impedance matching method, adjusts circuit parameters such as capacitance and inductance to achieve impedance matching. Passive impedance matching methods do not require additional active components, resulting in lower costs and higher stability [17]. However, passive impedance matching techniques often have lower efficiency and are typically suitable for short-distance wireless power transfer systems. In [18], an efficiency analysis and impedance matching design for strongly coupled wireless power transfer systems is proposed. The study adjusts resonant circuit parameters to achieve optimal efficiency transmission. However, passive impedance matching methods require adjustments based on different frequencies and distances, with limited effectiveness in long-distance transmission [19].

Based on a comprehensive comparison of active impedance matching and passive impedance matching, the following conclusions were drawn: Active impedance matching methods achieve impedance matching by adding active components, which enhance power transmission efficiency but increase system costs and complexity and are susceptible to stability issues [20]. Passive impedance matching methods achieve impedance matching by adjusting circuit component parameters and offering lower costs and higher stability; however, they typically exhibit lower efficiency and are generally suitable for short-distance transmission [21].

To achieve optimal efficiency transmission in wireless power transfer systems, researchers have proposed various control methods and optimization techniques. For example, ref. [22] proposed an integrated approach combining active impedance matching and adaptive power control to optimize WPT systems for implantable biomedical devices. The authors of [23] investigated efficiency optimization in wireless power transfer systems under different impedance matching topologies.

In this context, this paper presents a wireless power transfer efficiency optimization tracking method based on full current mode impedance matching. This method enables optimal efficiency transmission in wireless power transfer systems and real-time monitoring of system efficiency for timely adjustments and optimization. The proposed method utilizes impedance matching to control the output power of wireless power transfer systems. Modeling and analysis are conducted using a system state–space model for optimal efficiency transmission and efficiency detection. Furthermore, the feasibility and effectiveness of the proposed optimal efficiency tracking method and detection method were validated through simulation experiments conducted in MATLAB.

The contributions of this study can be summarized as follows: Firstly, a wireless power transfer efficiency optimization tracking method based on full current mode impedance matching is proposed, enabling optimal efficiency transmission in the WPT system. Secondly, the method extended the transmission efficiency of the detection mode in the wireless power transfer system. Finally, simulation experiments conducted in MATLAB validated the feasibility and effectiveness of the proposed method. This study’s research outcomes are significant for the development and application of wireless power transfer technology.

## 2. System Topology and Circuit Theory Analysis

### Topology Analysis of SS-Type Wireless Power Transfer System

The SS-type topology is a commonly used control structure in wireless power transfer systems. The SS-type controller is a state feedback controller based on the system’s state–space model, enabling precise control of system state and output. In wireless power transfer systems, the SS-type controller is primarily used to control the output power and achieve optimal efficiency transmission by controlling the magnitude and waveform of the output power.

Magnetic-coupling systems have four basic topologies: SS, SP, PS, and PP. In magnetic-coupling systems, the SP (Series–Parallel), PS (Parallel–Series), and PP (Parallel–Parallel) configurations complement the SS (Series–Series) topology as the four basic topologies.

SS (Synchronous–Synchronous): In an SS configuration, the magnetic coupling system’s input and output sides are synchronous, meaning that both the driving and driven components rotate at the same speed.

SP (Synchronous–Passive): In an SP configuration, the driving side is synchronous while the driven side is passive. The driving component rotates at a constant speed and transfers motion to the passive component through magnetic coupling.

PS (Passive–Synchronous): In a PS configuration, the driving side is passive while the driven side is synchronous. In this setup, the passive component receives motion from the synchronous component through magnetic coupling.

PP (Passive–Passive): In a PP configuration, both the driving and driven sides are passive. There is no synchronous component involved, and both sides rely on magnetic coupling for motion transfer without active driving force.

The SS-type system is widely used due to its inherent resonance frequency (ω), which remains unchanged with variations in equivalent resistance. In this study, a series-resonant WPT system with a single transmitter and receiver was investigated. A typical SS topology is shown in Figure 1, where Vac represents the AC voltage source; Lp and Ls denote the primary and secondary resonant inductors; and Cp and Cs represent the primary and secondary resonant capacitors, together forming the resonant network of the entire system. Rp and Rs represent the internal resistance of the primary and secondary coils, respectively, while RL represents the equivalent resistance of the system. Energy transfer between the coils is based on electromagnetic induction, where M denotes the mutual inductance between the coils, and the coupling coefficient can be calculated as $k=\frac{M}{\sqrt{L_{p}L_{s}}}$. The following is a schematic diagram of the SS-type wireless power transfer system topology:

Based on the above diagram, Kirchhoff’s Voltage Law (KVL) equations can be written as follows:(1)$$\left\{\begin{array}{c} v_{ac}=z_{1}I_{P}-j\omega MI_{S}\\ 0=z_{2}I_{S}-j\omega MI_{P} \end{array}\right.$$

By solving Equations (1) and (2) simultaneously, the input impedance Z_1_ and output impedance Z_2_ can be obtained:(2)$$\left\{\begin{array}{c} Z_{1}=R_{P}+j\omega L_{P}-\frac{1}{j\omega C_{P}}\\ Z_{2}=R_{S}+R_{L}+j\omega L_{S}-\frac{1}{j\omega C_{S}} \end{array}\right.$$

Rearranging the above equations, the input current I_p_ and output current I_s_ can be determined:(3)$$\left\{\begin{array}{c} I_{P}=\frac{Z_{2}V_{ac}}{Z_{1}Z_{2}+\omega^{2}M^{2}}\\ I_{S}=\frac{j\omega MV_{ac}}{Z_{1}Z_{2}+\omega^{2}M^{2}} \end{array}\right.$$

From the above equations, due to resonance, the imaginary part (j) can be disregarded. With $j\omega L_{P}-\frac{1}{j\omega C_{P}}=j\omega L_{S}-\frac{1}{j\omega C_{S}}=0$, the input power Pin can be calculated:(4)$$P_{in}=I_{P}V_{ac}=\frac{V_{ac}^{2}\left(R_{S}+R_{L}\right)}{R_{P}R_{S}+R_{P}R_{L}+\omega^{2}M^{2}}$$

Similarly, the output power Pout can be calculated:(5)$$P_{out}=I_{S}^{2}R_{L}=\frac{\omega^{2}M^{2}V_{ac}^{2}R_{L}}{{\left(R_{P}R_{S}+R_{P}R_{L}+\omega^{2}M^{2}\right)}^{2}}$$

The efficiency η can be determined as:(6)$$\eta_{ss}=\frac{P_{out}}{P_{in}}=\frac{\omega^{2}M^{2}R_{L}}{\left(R_{S}+R_{L}\right)\times \left(R_{P}R_{S}+R_{P}R_{L}+\omega^{2}M^{2}\right)}$$

After obtaining the output power P and efficiency η, the maximum power point and maximum efficiency point can be calculated:

Since $\frac{dP}{dR_{L}}=0$ and $\frac{d^{2}P}{d^{2}R_{L}}<0$, the maximum power point is derived as:(7)$$R_{L-Pmax}=\frac{\omega^{2}M^{2}}{R_{P}}+R_{S}$$

Since $\frac{d\eta }{dR_{L}}=0$ and $\frac{d^{2}\eta }{d^{2}R_{L}}<0$, the maximum efficiency point is derived as:(8)$$R_{L-\eta max}=\sqrt{R_{S}^{2}+R_{S}\frac{\omega^{2}M^{2}}{R_{P}}}$$

The above analysis can also be used to determine matching tracking points through the two-port network equivalent method of the science narrative. The two-port network equivalent method simplifies and analyzes complex circuit networks. It is based on the properties of two-port networks and involves finding an equivalent two-port network to replace the original circuit, simplifying the analysis process. As shown in Figure 2, an equivalent circuit is found on the left side of RL:

For convenience in subsequent calculations, let us define $Q_{P}=\frac{\omega L_{P}}{R_{P}},Q_{S}=\frac{\omega L_{S}}{R_{S}}$ and $Q=\sqrt{Q_{P}Q_{S}}$.

Therefore, the maximum output power point can be determined as:(9)$$R_{L-Pmax}=\frac{\omega^{2}M^{2}}{R_{P}}+R_{S}=R_{S}\left(1+{\left(KQ\right)}^{2}\right)$$

Similarly, the maximum efficiency point is provided by:(10)$$R_{L-\eta max}=\sqrt{R_{S}^{2}+R_{S}\frac{\omega^{2}M^{2}}{R_{P}}}=R_{S}\sqrt{1+{\left(KQ\right)}^{2}}$$

## 3. Optimal Efficiency Tracking Method in Full Current Mode

### 3.1. Secondary-Side Impedance Matching Topology and Analysis

Active impedance matching and passive impedance matching are key techniques for optimal efficiency tracking in wireless power transfer systems. Researchers have introduced active components or adjusted circuit parameters to achieve impedance matching, improving power transfer efficiency and system stability. However, different methods and techniques have variations in cost, efficiency, and stability, requiring selection and optimization based on specific application requirements.

Due to the rapid and wide-ranging load variations in the designed system, optimal efficiency tracking through passive impedance matching is difficult to achieve. Therefore, active impedance matching was adopted in this system. The secondary-side impedance matching topology is shown in Figure 3.

In this figure, Vin represents the AC voltage source; S1, S2, S3, and S4 are the four switch transistors; and Lp and Ls and Cp and Cs are the resonant inductors and capacitors on the primary and secondary sides, forming the resonant network of the entire system. Rp and Rs are the internal resistances of the primary and secondary coils, respectively, AIM (Active Impedance Matching) denotes the active impedance matching module, and RL represents the system’s equivalent resistance.

Active impedance matching, also known as active matching or active impedance control, is a technique using active electronic components to match the source or load impedance to the desired impedance. It involves active devices such as operational amplifiers (op-amps), transistors, or integrated circuits to actively adjust the impedance and achieve maximum power transfer or signal integrity.

In active impedance matching, a DC–DC equivalent converter is introduced, and the load resistance is transformed by controlling the duty cycle in the circuit. This technique allows control over the maximum power point and maximum efficiency point. Active impedance matching circuits have two forms: buck and boost circuits. The following analysis focuses on buck and boost circuits.

### 3.2. Analysis of Impedance Matching Circuit: Buck Circuit

Figure 4 shows buck circuit topology in active impedance matching. In this circuit, S represents the switch transistor, D represents the unidirectional diode, L represents the inductor, C represents the capacitor, RL represents the system’s equivalent resistance, Re represents the impedance value at the optimal efficiency point being tracked, and PWM represents pulse width modulation.

The control of the optimal efficiency tracking point is achieved by adjusting duty cycle D, which satisfies the following expression:(11)$$D=\sqrt{\frac{RL}{Re}}$$

Since duty cycle D must be greater than 1, RL must be greater than Re for the buck circuit to operate.

### 3.3. Analysis of Impedance Matching Circuit: Boost Circuit

Figure 5 depicts the boost circuit topology in active impedance matching. In this circuit, S represents the switch transistor, D represents the unidirectional diode, L represents the inductor, C represents the capacitor, RL represents the system’s equivalent resistance, Re represents the impedance value at the optimal efficiency point being tracked, and PWM represents pulse width modulation.

The control of the optimal efficiency tracking point is accomplished by adjusting duty cycle D, which satisfies the following expression:(12)$$D=1-\sqrt{\frac{RL}{Re}}$$

Since duty cycle D must be greater than 1, RL must be less than Re for the boost circuit to operate.

## 4. Extension of Traditional Impedance Tracking Topology

### 4.1. Overall Impedance Matching Topology

In this system, the efficiency tracking point is smaller than the equivalent resistance point, implying that RL is greater than Re. Therefore, the buck circuit’s active impedance matching form was adopted in this system. The overall impedance matching topology is shown in Figure 6. In this figure, Vd represents the AC voltage source; S1, S2, S3, and S4 are the four switch transistors; and Lp and Ls and Cp and Cs are the resonant inductors and capacitors on the primary and secondary sides, forming the resonant network of the entire system. Rp and Rs are the internal resistances of the primary and secondary coils; D1, D2, D3, and D4 are the unidirectional diodes; Cr is the inter-stage capacitor; Sb represents the buck circuit control switch transistor; Db is the unidirectional diode; Lb is the buck circuit inductor; Cb is the buck circuit capacitor; and Rb represents the system’s equivalent resistance.

The overall impedance matching topology, as shown in Figure 6, consists of an inverter circuit, resonant circuit, rectifier circuit, and buck circuit, without additional circuits. The inverter circuit is formed by four switch transistors. The resonant circuit consists of primary and secondary side resonant inductors and capacitors, as well as the internal resistances of the primary and secondary coils. The rectifier circuit consists of four unidirectional diodes. The buck circuit comprises switch transistors, diodes, an inductor, a capacitor, and output resistance.

The core control of this topology lies in the conduction and switching control of the buck circuit’s switch transistor Sb, which controls the buck circuit’s duty cycle D and, consequently, the continuity or discontinuity of the current flowing through the inductor Lb. This system enables the tracking and detection of the resistance value of the output resistance Rb in both the continuous conduction mode (CCM) and discontinuous conduction mode (DCM), expanding the range of impedance variations. We also analyzed the switching criteria between these two modes and the operation of CCM and DCM modes.

### 4.2. Switching Criteria between CCM and DCM Modes in the Buck Circuit

The buck circuit topology is shown separately in Figure 7, where Vb represents the input voltage after the rectifier circuit, Sb represents the control switch transistor of the buck circuit, Db represents the unidirectional diode, Lb represents the buck circuit inductor, Cb represents the buck circuit capacitor, and Rb represents the system’s equivalent resistance.

The buck circuit can operate in both continuous conduction mode (CCM) and discontinuous conduction mode (DCM). Based on the fundamental principles of the buck circuit, the current in the inductor does not drop to zero throughout the entire switching period in CCM. By contrast, in DCM, the current in the inductor drops to zero in each switching period. Mode switching is determined by the following equation:

According to the above equation, switching between CCM and DCM modes depends on duty cycle D controlled by Sb. Based on this equation, we modeled and analyzed the CCM and DCM modes separately.

### 4.3. Analysis of CCM Mode

Based on Equation (16), the relationship between the current iLb in inductor Lb and time t in CCM can be modeled and analyzed by adjusting the numerical parameters. Figure 8 shows the relationship between iLb and t in CCM, where iLb represents the current flowing through inductor Lb; ton is the conduction time of the switch transistor; and Tb is the end point of one period.

The mathematical analysis shown in Figure 8 reveals that the average voltage across the inductor within one period is zero, leading to the following equation:(13)$$\int_{0}^{t_{on}}\left(Vb-Vo\right)dt+\int_{t_{on}}^{T_{b}}\left(-Vo\right)dt=0$$

According to Equation (18), the output voltage of the buck circuit in CCM can be obtained as follows:(14)Vo=DVb

### 4.4. Analysis of DCM Mode

By adjusting the numerical parameters according to Equation (17), the relationship between the current iLb in inductor Lb and time t in DCM can be modeled and analyzed. Figure 9 illustrates the relationship between iLb and t in DCM, where iLb represents the current flowing through inductor Lb; ton is the conduction time of the switch transistor; t0 is the starting point of current interruption; and Tb is the end point of one period.

The mathematical analysis shown in Figure 9 reveals that the average voltage across the inductor within one period is zero, leading to the following equation:(15)$$\int_{0}^{t_{on}}\left(Vb-Vo\right)dt+\int_{t_{on}}^{t_{0}}\left(-Vo\right)dt=0$$

According to Equation (20), the output voltage of the buck circuit in DCM can be obtained as follows:(16)$$Vo=\frac{t_{on}}{t_{0}}Vb$$

Furthermore, considering that the average current of the inductor within one period is equal to the current on the load, we can derive the following equation:(17)$$t_{0}=\frac{t_{on}+\sqrt{t_{on}^{2}+\frac{{8L}_{b}T_{b}}{R_{b}}}}{2}$$

By combining Equations (21) and (22), we can determine the output voltage of the buck circuit in DCM as follows:(18)$$Vo=\frac{2}{1+\sqrt{1+\frac{{8L}_{b}}{R_{b}T_{b}D^{2}}}}Vb$$

## 5. System Simulation and Results Analysis

### 5.1. Mathematical Analysis of Model Establishment

According to the law of energy conservation, the relationships between the input resistance Rb-in and the output resistance Rb in continuous conduction mode (CCM) and discontinuous conduction mode (DCM) of the buck circuit can be obtained as follows:(19)$$R_{b}=\left\{\begin{array}{l} D^{2}R_{b-in}\;\;\;\;\;\;\;\;\;\;\;\;\;\;\;\;\;\;\;\;\;\;\;\;\;\;\;\;\;\;\;\;\;\;\;\;\;\;\;\;\;\;\;\;\;\;\;CCM\\ \frac{D^{4}R_{b-in}^{2}T_{b}^{2}-{4D}^{2}L_{b}R_{b-in}T_{b}+{4L}_{b}^{2}}{D^{4}R_{b-in}T_{b}^{2}}\;\;\;\;\;\;\;\;\;\;\;\;\;\;\;\;DCM \end{array}\right.$$

According to the law of energy conservation, when the input voltage is AC, the relationship between the AC equivalent impedance Ri of the buck circuit and the output impedance Rb in both continuous conduction mode (CCM) and discontinuous conduction mode (DCM) can be derived as follows:(20)$$R_{i}=\left\{\begin{array}{l} \frac{8}{\pi^{2}D^{2}}R_{b}\;\;\;\;\;\;\;\;\;\;\;\;\;\;\;\;\;\;\;\;\;\;\;\;\;\;\;\;\;\;\;\;\;\;\;\;\;\;\;\;\;\;\;\;\;\;\;\;\;\;\;\;CCM\\ \frac{4\left(D^{2}R_{b}T_{b}+{4L}_{b}+\sqrt{R_{b}T_{b}D^{2}\left(R_{b}T_{b}D^{2}+{8L}_{b}\right)}\right)}{\pi^{2}{D^{2}T}_{b}}\;\;\;\;\;DCM \end{array}\right.$$

A theoretical analysis of the two aforementioned equations reveals that in Continuous Conduction Mode (CCM), the relationship between the equivalent resistance and the output resistance solely depends on the square of the duty cycle $D^{2}$. Therefore, tracking the optimal efficiency point can be achieved by adjusting duty cycle D. Conversely, in Discontinuous Conduction Mode (DCM), factors influencing efficiency include inductance, time constants, and duty cycle interaction. By adjusting these parameters, tracking the optimal efficiency point can be achieved.

### 5.2. Model Control Block Diagram

Once the corresponding relationships between input resistance and output resistance in continuous and discontinuous conduction time are determined, the goals of impedance matching are clarified. The control block diagram of the impedance matching model is shown in Figure 10, where Vb represents the output voltage across Rb; Ib represents the output current through Rb; MD represents the multiplier and divider; Rb represents the output resistance; PWM represents the PWM controller; K represents the coupling coefficient; Q represents the expression defined in Section 2; R_(b-Pmax) represents the maximum power point; R_(b-ηmax) represents the maximum efficiency point; GD represents the gate driver module; MOS represents the metal-oxide-semiconductor field-effect transistor; D represents the duty cycle; CCM represents the continuous conduction time; DCM represents the discontinuous conduction time; IM represents impedance matching; and R_(b-out) represents the matched output resistance.

The model control block diagram analysis was as follows: In the first step, the output voltage and current through Rb were detected, multiplied, and divided to obtain the output resistance before impedance matching. In the second step, the obtained output resistance was input to the PWM controller, which adjusted the K and Q parameters derived from the maximum power and efficiency expressions in Section 2 to achieve maximum power or efficiency. The PWM controller controlled the GD gate driver module, which drove the MOS transistor. In the third step, the MOS transistor adjusted duty cycle D based on the expression derived from Section 3.2 to switch between CCM and DCM modes. Finally, the matched output resistance was obtained. Through these three steps, the model was effectively controlled. The model was simulated in MATLAB to validate the results.

### 5.3. Boundary Calculation of the Model

Based on the formulas derived from Section 4.1, the relationship between the duty cycle (Dc) and output resistance (Rb) in both CCM and DCM can be expressed using an intermediate variable, m, as follows:DC=(21)−6m−16m+13−I32m3−13m(22)m3+13m+13
(23)$$m=\frac{\sqrt[{3}]{\left(3I\sqrt{3}\sqrt{\left(-2R_{b-in}T+27L\right)+R_{b-in}T-27L}\right)R_{b-in}^{2}T^{2}}}{R_{b-in}T}$$

Boundary points are shown in Figure 11, where the *x*-axis represents the duty cycle (D); Rb represents the matched output resistance; and Dc represents the boundary point between CCM and DCM modes.

From Figure 11, we can observe that the system operates well in both modes, and the boundary point C is also clearly visible. Then, we adjusted the coupling coefficient parameter, K, and observed the operation of CCM and DCM under different K values.

### 5.4. Results Analysis of the Extended Model

The operation of CCM and DCM modes under different K values is shown in Figure 12 after adjusting the coupling coefficient K. The *x*-axis represents the duty cycle (D) and Rb represents the matched output resistance.

According to Figure 12, the system achieves good impedance matching and efficiency tracking. A larger K value allows for a wider impedance matching range. The system can operate and switch between CCM and DCM modes, achieving an extended tracking effect.

### 5.5. Experimental Data and Analysis

An impedance matching experimental platform based on the above analysis is shown in Figure 13, demonstrating the scientific narrative.

Table 1 presents experimental data obtained under different coupling coefficient K values, where Ib* and η* represent the experimental data, and Ib and η represent the simulation data. Ib represents the tracked current through the output resistance and η represents the transmission efficiency. These results are shown in the table below.

After consolidating the data, the current error graph and efficiency error graph for the seven coupling coefficient K values are shown in Figure 14.

A comparison of experimental and theoretical results reveals a certain degree of deviation. Among the reasons for this are the neglect of high-frequency radiation losses in theoretical calculations, the fact that the system’s self-resonant frequency is not always equal to the coil’s inherent frequency, and the precision with which distances are controlled between transmitting and receiving coils. However, both the tracked current and transmission efficiency have offsets within a range of 92–95%, indicating the system’s good tracking optimization.

To comprehensively assess the optimal efficiency tracking performance of the proposed system in this study, a comparative analysis was conducted with three relevant literature sources. Key areas of focus included impedance matching circuits, optimal efficiency tracking control methodologies, load response time, variations in coupling coefficients, changes in load resistance, and power levels. Our comparative findings are summarized in the table below. The comparative analysis of the literature is presented in Table 2.

## 6. Conclusions

The proposed method of impedance matching based on the full current mode for optimizing wireless power transfer efficiency has significant advantages in terms of energy transfer efficiency and tracking detection. This system uses an innovative approach to transferring energy to the target device. It achieves optimized energy transfer efficiency by controlling the conduction and shutdown of the switch transistor. In turn, the tracking effect expands and extends.

When using the full current mode impedance matching method, closely monitoring changes in transmission efficiency through real-time monitoring of system parameters such as input power, output power, and transmission distance is crucial. By analyzing and optimizing these parameters, the system can achieve maximum efficiency and optimal energy utilization results. The impedance matching method for wireless power transfer described in this paper, based on full current mode, achieved excellent efficiency tracking in both modes. Whether tracking current or efficiency, the matching degree was controlled within a range of 92–95%.

Furthermore, the impedance matching method based on the full current mode for optimizing wireless power transfer efficiency has great potential for further expansion. By modifying the system’s circuit parameters and structure, transmission efficiency and distance can be further optimized. This method can be extended to various applications, including wearable and IoT devices. Therefore, the application and optimization of this system should be explored and studied to achieve broader applications and more efficient energy utilization.

In summary, the impedance matching method based on the full current mode for optimizing wireless power transfer efficiency has significant advantages in terms of energy transfer efficiency, and tracking detection. It also has great expansion potential. In future research, we will continue to explore and optimize this system to achieve better results and broader applications.

## Figures and Tables

**Figure 1 sensors-24-02917-f001:**
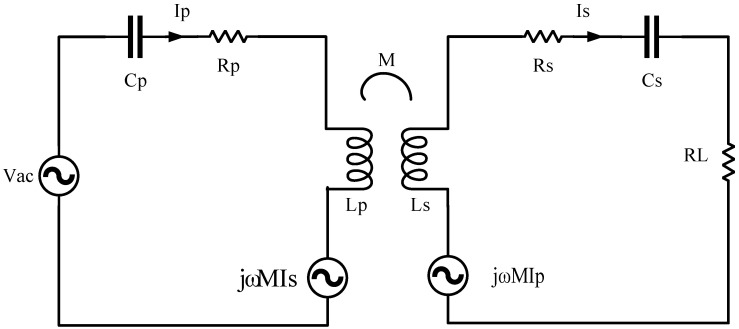
SS-type topology.

**Figure 2 sensors-24-02917-f002:**
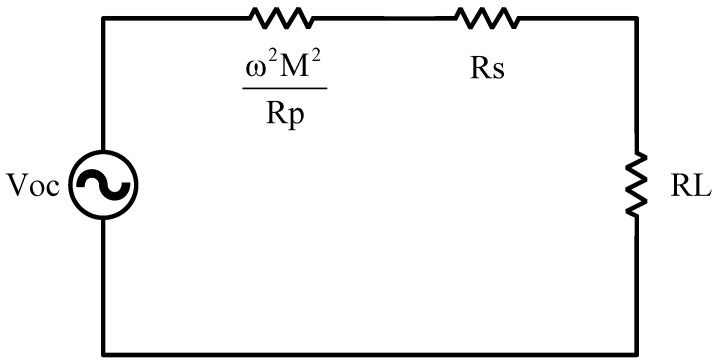
Two-port equivalent topology.

**Figure 3 sensors-24-02917-f003:**
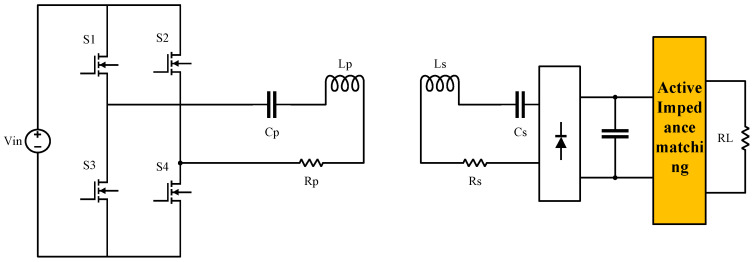
Secondary-side impedance matching topology.

**Figure 4 sensors-24-02917-f004:**
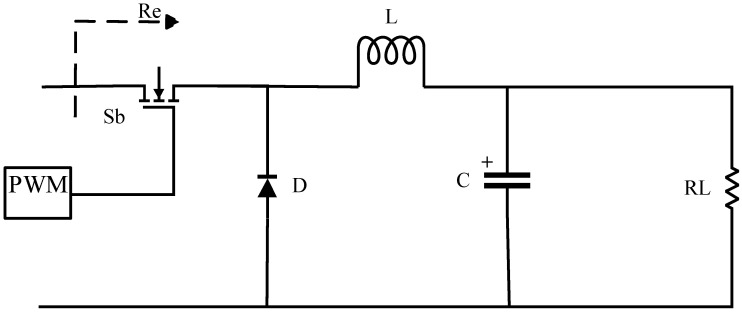
Impedance matching buck circuit.

**Figure 5 sensors-24-02917-f005:**
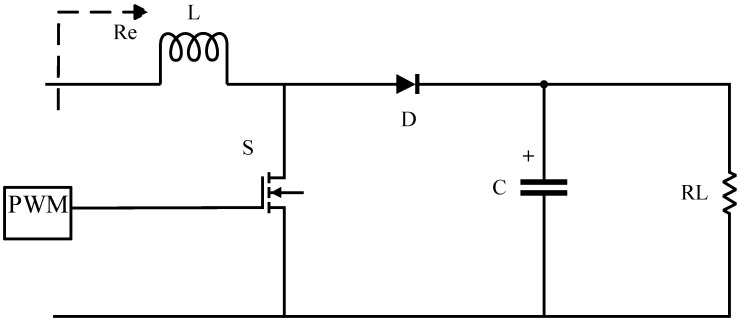
Impedance matching boost circuit.

**Figure 6 sensors-24-02917-f006:**
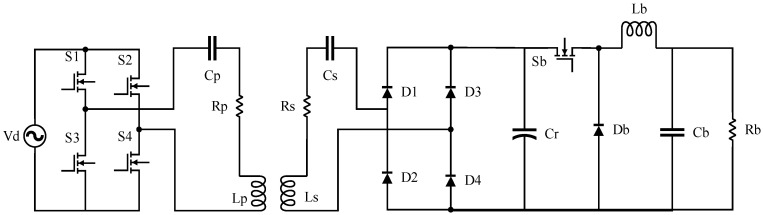
Overall impedance matching topology.

**Figure 7 sensors-24-02917-f007:**
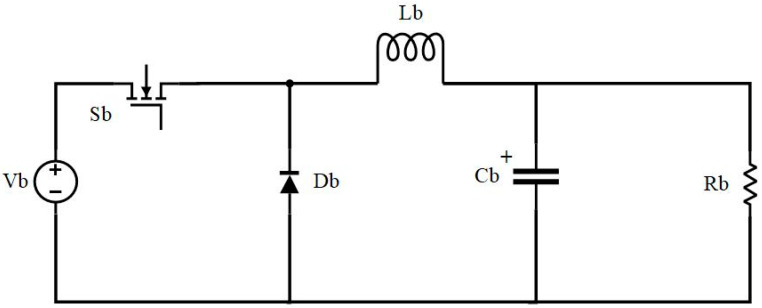
Buck circuit topology.

**Figure 8 sensors-24-02917-f008:**
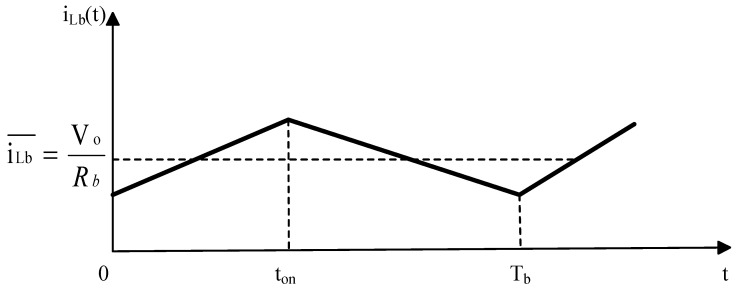
CCM Mode.

**Figure 9 sensors-24-02917-f009:**
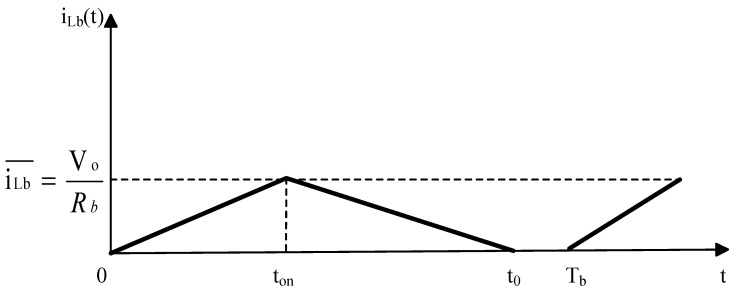
DCM Mode.

**Figure 10 sensors-24-02917-f010:**
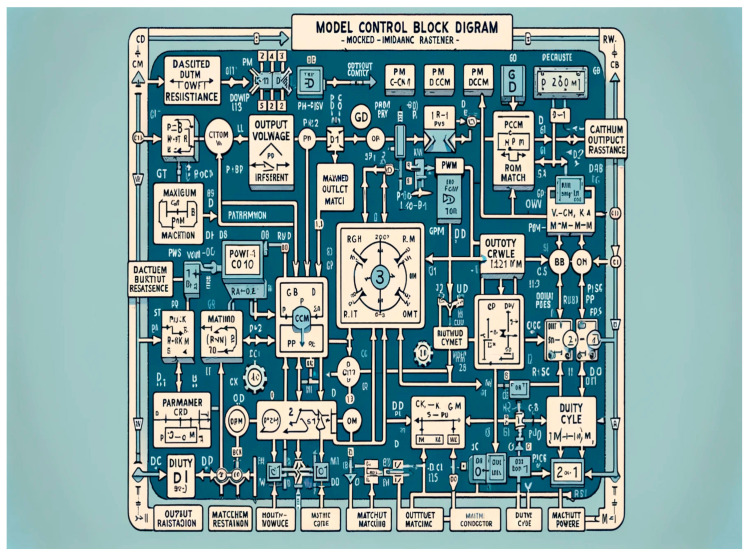
Model control structure diagram.

**Figure 11 sensors-24-02917-f011:**
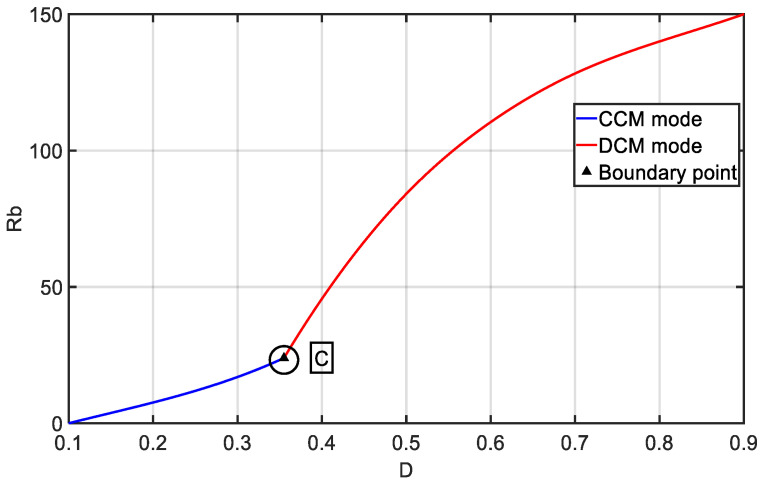
Boundary Calculation.

**Figure 12 sensors-24-02917-f012:**
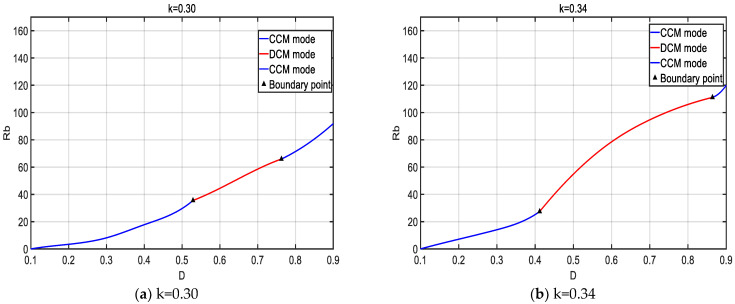
Impedance matching under different K inputs.

**Figure 13 sensors-24-02917-f013:**
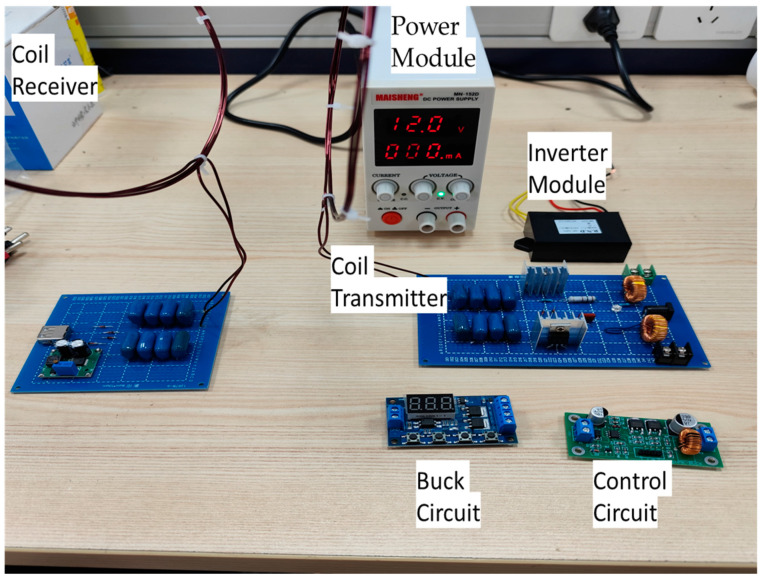
Impedance matching experimental platform.

**Figure 14 sensors-24-02917-f014:**
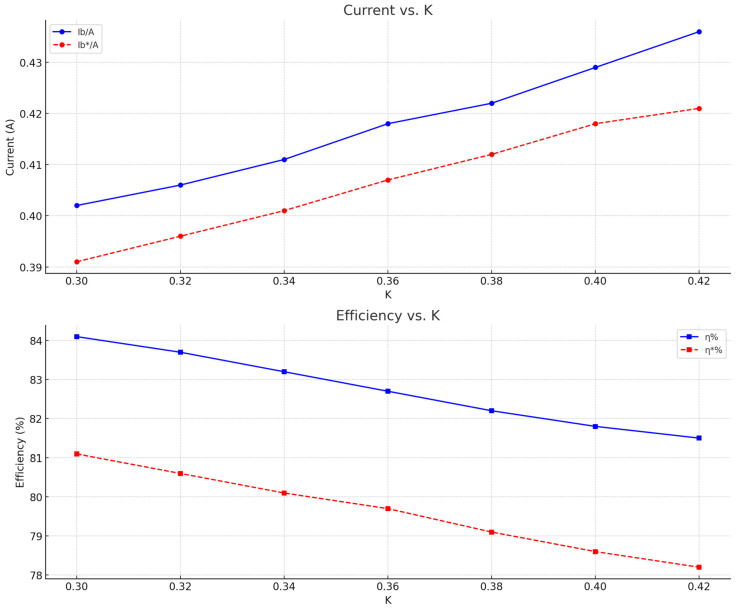
Current error and Efficiency error graph.

**Table 1 sensors-24-02917-t001:** Experimental results.

K	I_b_*/A	I_b_/A	η*	η
0.30	0.391	0.402	0.811	0.841
0.32	0.396	0.406	0.806	0.837
0.34	0.401	0.411	0.801	0.832
0.36	0.407	0.418	0.797	0.827
0.38	0.412	0.422	0.791	0.822
0.40	0.418	0.429	0.786	0.818
0.42	0.421	0.436	0.782	0.815

**Table 2 sensors-24-02917-t002:** Comparative analysis of the literature.

Literature	Impedance Matching Circuits	Optimal Efficiency Tracking Control Methods	Load Response Time		Overall Efficiency
			Variations in Coupling Coefficients	Changes in Load Resistance	
This paper	Both modes employ buck conversion circuits	Adjusting the converter duty cycle	Approximately 170 ms	Approximately 160 ms	Approximately 80.2%
[24]	With a buck–boost conversion circuit in DCM mode	Adjusting the converter duty cycle in DCM mode	Approximately 180 ms	Approximately 150 ms	Approximately 80%
[25]	With a buck–boost conversion circuit in CCM mode	Adjusting the converter duty cycle in CCM mode	Approximately 800 ms	Approximately 600 ms	Approximately 80%
[26]	Along with a T-type passive impedance matching circuit	The tuning of the passive impedance matching network’s inductance and capacitance values	Approximately 200 ms	Approximately 200 ms	Approximately 80%

## Data Availability

Not applicable.

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
