# Peer review of "Wireless Power Transfer Efficiency Optimization Tracking Method Based on Full Current Mode Impedance Matching"

_sensors, 2024, doi:10.3390/s24092917_

Round 1

Reviewer 1 Report

Comments and Suggestions for Authors

In Section 2; SS-Type Wireless Power Transfer System define…. Define SS as series-series …

In page 3, 2nd para “Currently, magnetic-coupling systems mainly consist of four basic topologies: SS, SP, PS, and PP.” define SP, PS and PP.

Equations (3) and (4) are incorrect, check the signs. Moreover, Ip should ne Lp and Is should be Ls

Equation (7) is incorrect

Equation (9) is incorrect; The Power Transfer Efficiency = Pout/Pin

The equations (1) to (13) should be revised carefully.

Comments on the Quality of English Language

None

Author Response

Thank you so much for taking the time to review our paper and provide your valuable feedback and questions. Your expert insights helped refine our research and improve the quality of this paper. We made the modifications based on your suggestions and hope they meet your expectations. Thank you again for your guidance and support!

Reviewer 2 Report

Comments and Suggestions for Authors

This paper proposed a tracking method based on full current mode impedance matching for optimizing the efficiency of wireless power transfer. There are certain innovation points, but the overall quality of the manuscript is medium. Hence, I suggest giving a minor revision. Detailed comments can be found as follows: 

1. The author did not fully summarize the limitations and problems of WPT system control methods. Existing control methods for WPT systems have certain limitations and issues, such as low transmission efficiency, low interoperability (https://doi.org/10.3390/en16041653), low security (https://doi.org/10.1109/ACCESS.2023.3332470), short transmission distance, and high system costs.

2. The format of letters and subscripts in formulas and main text should be standardized. Now many different styles are mixing in the whole manuscript.

3. The indentation of the formula needs to be modified for improved readability.

4. The experimental platform in Fig. 13 requires improvement. Excess items should be removed, and the system composition should be clearly labelled.

5. The efficiency units in Table 1 and Fig. 15 are %. The efficiency value should not be around 0.8%.

6. The number of test points in the experiment is relatively small. It is recommended to supplement the number of test points to provide a more comprehensive analysis.

Author Response

(The authors gave the same response as above.)

Reviewer 3 Report

Comments and Suggestions for Authors

This paper presents the adding buck circuit to improve the efficiency system. However, I have some concerns about this manuscript.

 1.  Please analyze the difference between buck and boost circuits in your impedance matching circuit. Please clearly explain why you chose the buck circuit. 

2. How does the buck circuit based on full current mode impedance matching optimize the efficiency of wireless power transfer systems?

3. How were the simulation results of the proposed method validated in the MATLAB software environment? You need to compare with the state of the art.

4. Please provide the comparison table.

Author Response

(The authors gave the same response as above.)

Reviewer 4 Report

Comments and Suggestions for Authors

The work was carried out on an urgent topic related to wireless power transmission. This area is developing rapidly. The authors propose a method for tracking the optimization of the efficiency of wireless energy transmission based on impedance matching in full current mode. A number of both theoretical and practical results were obtained, the MatLab software package was applied and a simulation model was created. There are some comments/suggestions about the work: 1) The list of references is limited to the overview in the introduction, there are no references in further sections. It is hard to believe that all other theoretical information is original and obtained by the authors for the first time in this work. 2) In the list of references, for a single case, there are no references to the works of 2023, although the topic is very relevant and there are a large number of recent works. 3) Figures 1-7 show block diagrams of various topologies of electrical circuits and there is no model developed in MatLab anywhere, as stated by the authors. 4) In conclusion, information is provided on the various advantages of the method developed by the authors, but in what (any numerical values) they are expressed not specified. This needs to be fixed.

Author Response

(The authors gave the same response as above.)

Round 2

Reviewer 1 Report

Comments and Suggestions for Authors

After considering the comments by the authors, I can see that the manuscript became more matured and suitable for publishing.
Just one comment; in Table 2, put the results of "This Paper" at the end, i.e., after the results of [21]. [22] and [23].

.